DATA RELEASE

# The first genome assembly of the amphibian nematode parasite (*Aplectana chamaeleonis*)

Lei Han[1,2,†], Tianlu Liu[1,2,†], Fengping He[3,*] and Zhijun Hou[1,2,*]

1 Laboratory of Vector-Borne Diseases and Pathogens Ecology, College of Wildlife and Protected Area, Northeast Forestry University, Harbin 150040, China
2 BGI Life Science Joint Research Center, Northeast Forestry University, China
3 College of Veterinary Medicine, Yunnan Agricultural University, Kunming, China

## ABSTRACT

Cosmocercoid nematodes are common parasites of the digestive tract of amphibians. Genomic resources are important for understanding the evolution of a species and the molecular mechanisms of parasite adaptation. So far, no genome resource of Cosmocercoid has been reported. In 2020, a massive Cosmocercoid infection was found in the small intestine of a toad, causing severe intestinal blockage. We morphologically identified this parasite as *A. chamaeleonis*. Here, we report the first *A. chamaeleonis* genome with a genome size of 1.04 Gb. The repeat content of this *A. chamaeleonis* genome is 72.45%, and the total length is 751 Mb. This resource is fundamental for understanding the evolution of Cosmocercoid and provides the molecular basis for Cosmocercoid infection and control.

**Subjects** Genetics and Genomics, Zoology, Molecular Infection Biology

**Submitted:** 21 September 2022

\* Corresponding authors. E-mail: houzhijundb@163.com; hefengping@outlook.com

† Contributed equally.

Preprint submitted at https://doi.org/10.1101/2023.03.20.533390

## INTRODUCTION

Amphibians are widely distributed across the world, with a high diversity of species. More than 6,000 amphibian species exist worldwide, significantly contributing to the global biodiversity (http://www.amphibiachina.org/). Nematodes can interfere with the fitness of their amphibian hosts and ultimately affect their survival [1]. They can also facilitate adaptions in their host physiology, acting as a major force to promote their coevolution. Parasitic infections are highly prevalent in amphibians, but few systematic studies have been conducted on this topic [2]. The order Ascaridomorpha, a typical representative of large parasitic nematodes [3], contains five families: Ascaridoidea, Cosmocercoidea, Heterakoidea, Seuratoidea and Subuluroidea. Only Ascaridoidea has genomic data, and most of its genera have mammalian hosts. The genomic data of amphibian parasites from all five families are deficient. Within Ascaridomorpha, Cosmocercoidea is the most abundant family of amphibian roundworms [4]. The genus *Aplectana* (Cosmocercoidea: Cosmocercidae), which includes *Aplectana hylae* [5], *Aplectana macintoshii*, *Aplectana hainanensis* [6], *Aplectana paucipapillosa* [6], *Aplectana xishuangbannaensis* [7] and *Aplectana chamaeleonis* [1], includes common parasites of amphibians. Both nematodes and amphibians are important components of the ecosystem. The lack of molecular resources for amphibian parasites made it impossible to study their adaptability and evolutionary history, thus hindering the development of related fields. At present, research on *Aplectana* mainly focuses on its morphological identification. However, this genus is

difficult to distinguish only through morphology, as this parameter is affected by the developmental time and individual differences. Therefore, understanding the genus *Aplectana* simply through morphological identification is not enough.

In the present study, we report the first highly complete *A. chamaeleonis* genome, with a genome size of 1.04 Gb. Its repeat element content reaches 72.45%, providing new evidence for understanding the relationship between repeat elements and genome size in Ascaridomorpha species. Furthermore, as this is the first genome of a Cosmocercidae species, it enhances our understanding of the evolution of Cosmocercidae species and their adaptive molecular mechanisms to amphibian hosts.

## MAIN CONTENT

### Context

We present a highly-complete genome assembly of *A. chamaeleonis* (Figure 1, NCBI:txid2696335), providing a valuable resource for evolutionary biology, ecology and phylogenetics. The genome size is 1.04 Gb (Table 1). The average sequence length is 496 kb, and the N50 length is 1.08 Mb. The maximum length is 8.07 Mb, and the minimum is 34 kb. Our *A. chamaeleonis* genome has a GC content of 45%, and its N50 is longer than in other nematodes at the scaffold level while it is relatively shorter than in some of them at the chromosome scale. Additionally, the integrity of the genome was assessed at 76.9% using Benchmarking Universal Single-Copy Orthologs (BUSCO, RRID:SCR_015008). The characteristics of the genome sequence showed that the genome is large and has high integrity. Blobtools (RRID:SCR_017618) was used for genomic quality control and taxonomic partitioning. The results showed that 91% of the sequences aligned to Nematoda (1898/2088) and 7% to Arthropoda (122/2088). This will be an invaluable resource for understanding amphibian parasites.

According to the reported total repeat length and proportion of the Ascaridomorpha species, the content of the repeat elements in nematodes varies significantly, ranging from 3.92% to 45.25% [8]; in species with larger genomes, the content of repeat elements occupies a large proportion. There were significant differences in the total length of the genome and proportion of repeat content among different nematodes, which varied from 87 Mb to 751 Mb and from 8.34% to 72.45%, respectively (Table 1) [9]. In *A. chamaeleonis*, the total length of the genome is 1,036,852,746 bp, and the content of repetitive elements in the genome reaches a staggering 72.45%, with a total length of 751 Mb (Tables 1–3). We counted the content of the various repeating elements: unknown types of repeating elements account for 51%, whereas LINE and DNA account for 10% and 8%, respectively (Figure 2). These findings confirm that the large number of repeated sequences is one of the leading causes for such a large genome.

A total of 12,887 functional genes were annotated. Additionally, all genes were annotated with KEGG (RRID:SCR_012773) and found to be primarily represented in pathways such as 'Environmental Information Processing and Metabolism', indicating the role of signal transduction-related genes in *A. chamaeleonis* (Figure 3). In addition, all twelve metabolic pathways were found in the enrichment analysis of the *A. chamaeleonis* genes. The most enriched metabolic pathway was 'Carbohydrate metabolism', while the least enriched one was 'Biosynthesis of other secondary metabolites'. According to the annotation and enrichment using KEGG and KOG databases, 'Signal transduction' and 'Signal transduction



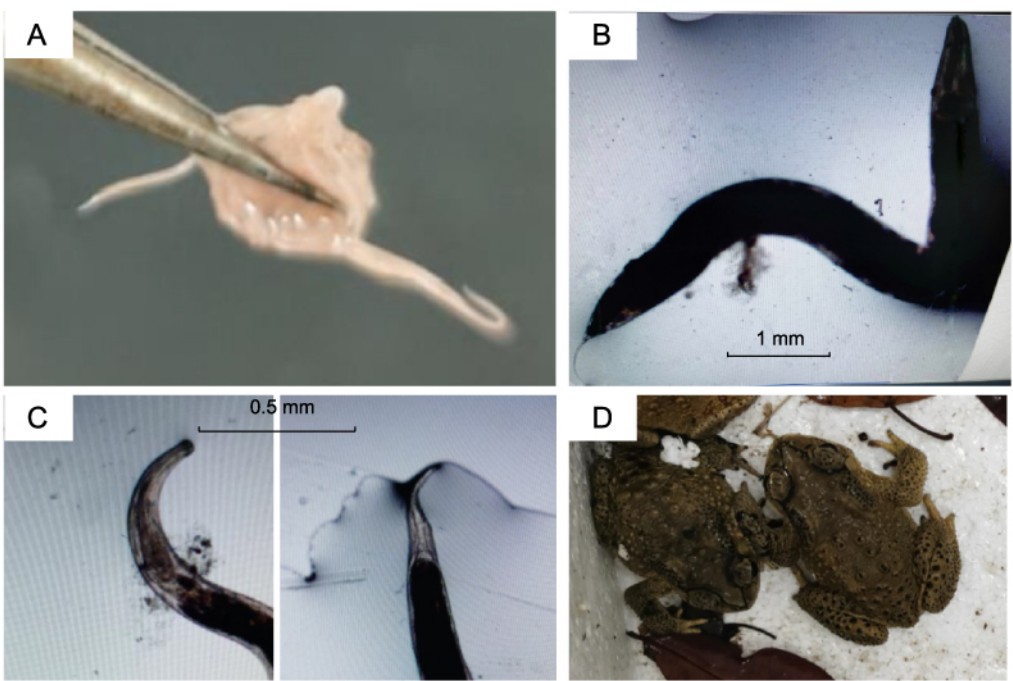

**Figure 1. Morphological characteristics of *A. chamaeleonis*.**
(a) Sample collection of *A. chamaeleonis*. (b) Microscopic observation of morphological characteristics. (c) Head (left) and tail (right) features of the *A. chamaeleonis*. (d) The *A. chamaeleonis* host toad (*Bufo pageoti*).

**Table 1.** Comparison of genomic information between *A. chamaeleonis* (our data) and other nematodes.

| | Total number | Total length (bp) | Average length (bp) | N50 Length (bp) | GC content (%) | Repeat content (%) |
|---|---|---|---|---|---|---|
| *A. chamaeleonis* | 2,088 | 1,036,852,746 | 496,577 | 1,075,597 | 45.82 | 72.45 |
| *Ascaris suum* | 415 | 298,028,455 | 718,140 | 4,646,302 | 37.79 | 8.78 |
| *Toxocara canis* | 22,857 | 317,115,901 | 13,874 | 375,067 | 39.95 | 9.16 |
| *Anisakis simplex* | 42,005 | 126,869,778 | 3,020 | 9,290 | 36.74 | 8.34 |
| *Brugia malayi* | 196 | 87,155,713 | 444,672 | 14,214,749 | 28.42 | 15 |
| *Caenorhabditis elegans* | 6 | 100,272,607 | 16,712,101 | 17,493,829 | 35.44 | 13 |

**Table 2.** Statistics for the repetitive sequences identified in the *A. chamaeleonis* genome, classified according to the biological category.

| Type | Length (bp) | % in genome |
|---|---|---|
| DNA | 83,137,834 | 8.02 |
| LINE | 109,657,137 | 10.58 |
| SINE | 225,965 | 0.02 |
| LTR | 22,133,402 | 2.13 |
| Other | 0 | 0 |
| Satellite | 1,105,518 | 0.11 |
| Simple_repeat | 1,014,078 | 0.10 |
| Unknown | 536,951,622 | 51.79 |
| Total | 741,597,934 | 71.52 |

mechanisms' accounted for large proportions of genes: 681 and 1,105, respectively. This finding may be due to the amphibious life of the amphibian hosts of *A. chamaeleonis*.

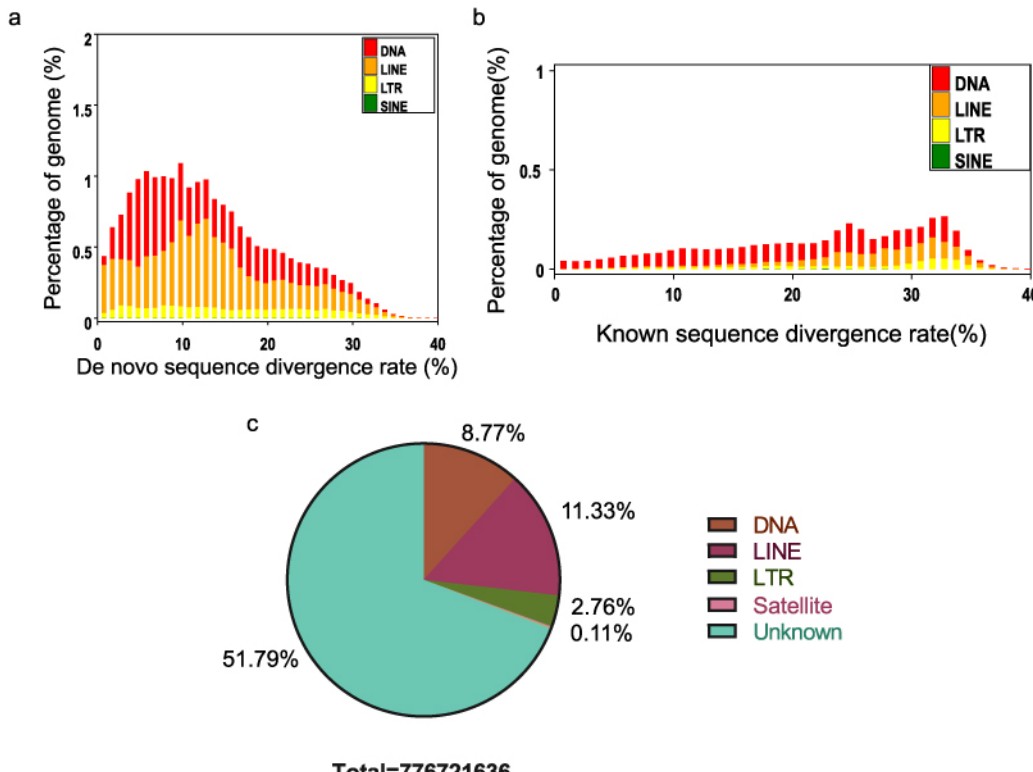

**Figure 2.** Distribution of transposable elements (TEs) in the *A. chamaeleonis* genome. The TEs include DNA transposons (DNA) and RNA transposons (i.e., DNAs, LINEs, LTRs, and SINEs).
(a) *De novo* sequence divergence rate distribution. (b) Known sequence divergence rate distribution. (c) Proportion and distribution of repeating elements.

**Table 3.** Summary of TEs in the *A. chamaeleonis* genome.

| Type | Repbase TEs | | TE proteins | | *De novo* | | Combined TEs | |
|---|---|---|---|---|---|---|---|---|
| | Length (bp) | % in genome | Length (bp) | % in genome | Length (bp) | % in genome | Length (bp) | % in genome |
| DNA | 21,977,537 | 2.12 | 7,492,953 | 0.72 | 83,137,834 | 8.01 | 90,920,889 | 8.77 |
| LINE | 11,773,738 | 1.14 | 79,496,840 | 7.67 | 109,657,137 | 10.58 | 117,472,126 | 11.33 |
| SINE | 257,558 | 0.02 | 0 | 0 | 225,965 | 0.02 | 431,378 | 0.04 |
| LTR | 5,868,341 | 0.57 | 8,431,837 | 0.81 | 22,133,402 | 2.13 | 28,619,319 | 2.76 |
| Other | 4,364 | 0.01 | 0 | 0 | 0 | 0 | 4,364 | 0.01 |
| Unknown | 0 | 0 | 0 | 0 | 536,951,622 | 51.79 | 536,951,622 | 51.79 |
| Total | 37,589,750 | 3.63 | 95,411,725 | 9.20 | 739,478,338 | 71.32 | 751,195,390 | 72.45 |

According to the demographic history scale of *A. chamaeleonis* (Figure 4), the population size of *A. chamaeleonis* gradually increased between 200,000 and 100,000 years ago. Then, during the last glacial maximum, the population of *A. chamaeleonis* gradually decreased, which may be linked to the decline of its host population during the same period [10].

The infection rate of the Cosmocercidae species in amphibians is high, but no genome information is available for any Cosmocercidae species. The genomes of *A. chamaeleonis* assembled in this study are an important resource for studying amphibian parasites. In particular, it may improve our understanding of the evolution of amphibian parasites and the molecular basis of the genome for adapting to amphibian hosts.

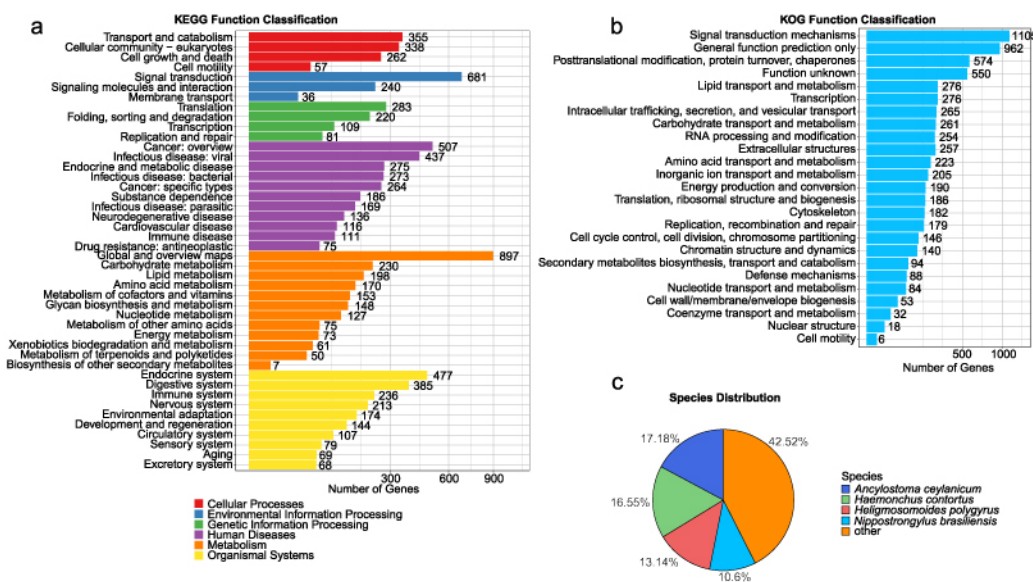

**Figure 3.** **Gene annotation information of *A. chamaeleonis*.**
(a) KEGG enrichment of *A. chamaeleonis*. (b) KOG enrichment of *A. chamaeleonis*. (c) Homologous species annotation distribution of *A. chamaeleonis*.

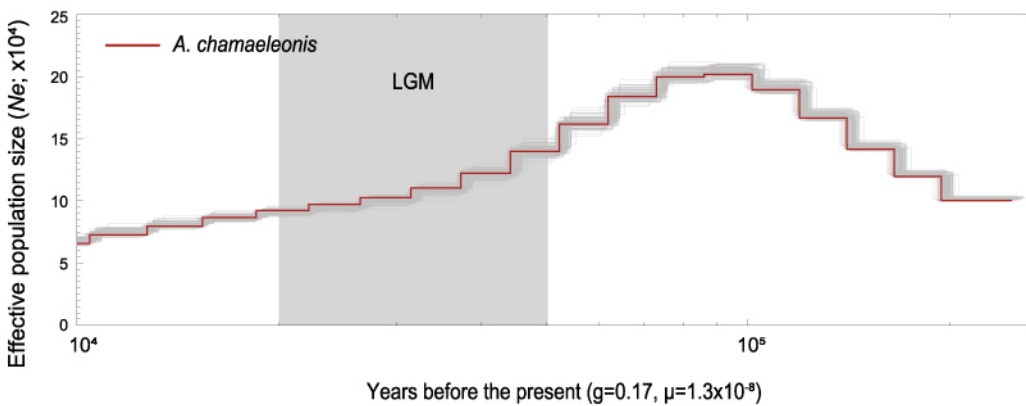

**Figure 4.** **Demographic history of *A. chamaeleonis*.**

## METHODS

### Sample collection and sequencing

*A. chamaeleonis* was collected from a *Bufo pageoti* infected with *A. chamaeleonis* in Shenzhen, China. All samples were thoroughly cleaned with sterile physiological saline (37 °C), quickly frozen, transported on dry ice, and kept at −80 °C until further use. By using the microscope, morphological identification was carried out (Olympus). All experimental designs and nematode handling were approved by the Institutional Animal Care and Use Committee of Northeast Forestry University. Sodium dodecyl sulphate/proteinase K digestion, phenol-chloroform extraction, and ethanol precipitation were used to isolate whole genomic DNA [11]. The DNA quantity was estimated using a Qubit fluorometer with

the dsDNA high-sensitivity kit (Invitrogen) and using agarose gel (1.0%) electrophoresis. Genomic DNA was purified for long-read library preparation according to the manufacturer's instructions of the Nanopore platform, followed by long-read sequencing.

## Genome assembly, duplicate purging

The Nanopore long reads were assembled with NextDenovo software (v2.0-beta.1; https://github.com/Nextomics/NextDenovo). Then NextPolish (v1.0.5) [12] was used to conduct a second round of correction and a third round of polishing for this assembly using the Whole Genome Sequencing (WGS) data. We used diamond (v0.9.10; RRID:SCR_016071) to blast the genome against the NCBI Non-Redundant Protein Sequence Database (NR) database. We then deleted the scaffolds that blasted to bacteria (such as *Escherichia coli* and *Lactococcus lactis*) and generated the clean genome. To get a haploid representation of the genome, duplicates were purged from the genome using the Purge_Dups pipeline (RRID:SCR_021173) [13]. To evaluate the quality of the genomes, a new software called blobtools was used for genomic quality control and taxonomic partitioning. The completeness of the genome was evaluated using the sets of BUSCO (v5.2.2) with genome mode and lineage data from nematode odb9 and eukaryote odb9, respectively [14].

Next, we used Tandem Repeats Finder [15], LTR_FINDER (RRID:SCR_015247) [16] and RepeatModeler (v2.0.1; RRID:SCR_015027). RepeatMasker (RRID:SCR_012954) [17] and RepeatProteinMask [18] were used to search the genome sequences for known repeat elements. The BRAKER2 pipeline (RRID:SCR_018964) [19] was used to perform gene prediction. Then we aligned the gene sets against several known databases, including SwissProt [20], TrEMBL [20], KEGG [21], KOG [22], COG [22], GO [23] and NR [24]. In addition, we used the pairwise sequentially Markovian coalescent model to estimate the effective population size of *A. chamaeleonis* within the last million years.

## Demographic history of *A. chamaeleonis*

We inferred the demographic history of *A. chamaeleonis*. The generation we used was 0.17 years per generation, and the mutation rate was $9 \times 10^{-9}$ single nucleotide mutations per site per generation on average [8]. We also used 100 bootstrap replicates to estimate the demographic history.

## DATA AVAILABILITY

The data that support the findings of this study have been deposited into the CNGB Sequence Archive (CNSA) [25] of the China National GeneBank Database (CNGBdb) [26] with the accession number CNP0003496, and in NCBI under the biosample number PRJNA895947. Additional data is also available in the GigaDB repository [27].

## DECLARATIONS

## List of abbreviations

BUSCO, Benchmarking Universal Single-Copy Orthologs; COG, Clusters of Orthologous Groups of Proteins; KEGG, Kyoto Encyclopedia of Genes and Genomes; KOG, Clusters of Orthologous Groups for Eukaryotic Complete Genomes; LINE, Long Interspersed Nuclear Elements; LTR, Long Terminal Repeats; NR, Non-Redundant Protein Sequence Database; SINE, Short Interspersed Nuclear Elements; TE, Transposable elements.

## Ethics approval and consent to participate

All experimental designs and nematode handling were approved by the Institutional Animal Care and Use Committee of Northeast Forestry University and performed in accordance with the laboratory of Entomopathogenic Diseases and Pathogen Ecology.

## Competing Interests

The authors declare no conflict of financial interests.

## Funding

This work was supported by the grants of Surveillance of Wildlife Diseases from the State Forestry Administration of China.

## Author contribution

ZH designed and initiated the project. LH collected the samples. LH and TL performed the DNA extraction, library construction and data analysis. LH and TL wrote the manuscript. All authors have read and approved the final manuscript.

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
