## [Reviewer Report]

Comments on revised manuscriptThe author revised the paper as I concerned in the report and the paper could be accepted now.

---

## [Reviewer Report]

Comments on revised manuscriptOverall, the writing has been improved in several places and is somewhat clearer than in the previous draft. These changes are mostly related to the minor concerns raised. However, many questions related to the broader impact of this research and how the new genome compares to other nematode species remain unanswered. The following comments were largely ignored.  1. The reasoning behind why this research was undertaken is not clear. 2. What is the ecological or agricultural and economic impact of the species? How would the genome provide a better understanding of this species? 3. More specific information is also needed to better understand the genome. How many chromosomes does this species have? Is there any cytology to help answer this question? Any notion of sex chromosome vs. autosome? 4. This genome is much bigger than most of the assembled parasitic nematodes. The author did not make an effort to explain what might contribute to this. 5. Overall, there is a lack of in-depth data analysis and comparison between this genome and many other available nematode genomes. How do these results compare to related species? 6. About the overall presentation and organization of the manuscript, the context is often lacking from results. Another round of general proofreading needs to be done for grammar, punctuation, capitalization, italics, etc. – see below for additional specific examples. The authors, not the reviewers, need to make a concerted effort to read and proofread their own manuscript.  In addition to the big picture points raised above, several other issues that were either brought up last time or are new and need to be addressed: 1. Not sure Table 1 is present the right way. The columns and rows should be reversed, I think. If so, there will be only one column - do you still need a table? 2. “Through the characteristics of the genome sequence, it shows that the genome is a highly continuous genome.” Unclear. The authors mentioned that they have fixed this in their response to the reviewers, but no change was seen in the updated manuscript. 3. “The generation we used was 0.17, and the mutation rate was 9×10-9 [8].” These numbers need units after them. Again, this was addressed in the response but not written out or clarified in the revised text. 4. “In addition, the enrichment of A. chamaeleonis genes in all metabolic pathways was found in twelve metabolic pathways.” Not sure what the authors were trying to say about the all or 12 pathways. Still confusing. 5. Photographs of the worms are still lacking scale bars. 6. Make sure that all genus and species names are italicized (in body text and in Fig.3). 7. Make section heading format is consistent (check capitalization). 8. “The results showed that 91 % of the sequences were compared to Arthropoda (1898/2088) and 7 % were compared to Arthropoda (122/2088).” Both of these say Arthropoda - is that a mistake? Also "compared to" is not the correct word, maybe "similar to"? 9. LGM acronym is defined after the second use of "last glacial period", should appear after the first use. Also, LGM stands for last glacial maximum, not period. This should be corrected.

---

## [Reviewer Report]

Reviewer name and names of any other individual's who aided in reviewer Xuanmin GuangDo you understand and agree to our policy of having open and named reviews, and having your review included with the published papers. (If no, please inform the editor that you cannot review this manuscript.)YesIs the language of sufficient quality?NoPlease add additional comments on language quality to clarify if needed
the language is so bad, and should ask an English speaker for revise the paper.Are all data available and do they match the descriptions in the paper? YesAdditional CommentsAre the data and metadata consistent with relevant minimum information or reporting standards? See GigaDB checklists for examples <a href="http://gigadb.org/site/guide" target="_blank">http://gigadb.org/site/guide</a>YesAdditional CommentsIs the data acquisition clear, complete and methodologically sound?YesAdditional CommentsIs there sufficient detail in the methods and data-processing steps to allow reproduction?YesAdditional CommentsIs there sufficient data validation and statistical analyses of data quality? NoAdditional CommentsIs the validation suitable for this type of data?YesAdditional CommentsIs there sufficient information for others to reuse this dataset or integrate it with other data?YesAdditional CommentsAny Additional Overall Comments to the AuthorHan et al. had carried out genome assembly of Aplectana chamaeleonis, analysised the genome’s repeat content and annotated the genome. They descripted the geneset’s function and done a PSMC analysis. The genome is a key source for research, but there are so many mistakes in the manuscript, I suggest the author to revies the manuscript carefully and the grama and content should be re-organized.  Some suggestions have been listed below: 1. In the context part, the first two sentence lacks continuity in logic, please change them. 2. The author didn’t mention which sequence platform they had used in the context, I think this should be added. 3. The average sequence length in the table is 496kbp, but the author it as 496Mbp , this is a mistake. 4. In table 1, why there aren’t any gaps in the scaffold genome? 5. The author said that “This suggests that the significant expansion of repeating elements is an important manifestation of species differences”. Its unreasonable to get this conclusion only based your genome repeat analysis. 6. In the text they claim that 12887 function gene had annotated, I want to know how much gene they have annotated? Please add this in the manuscript. 7. Too much decimal place has used in the Table2RecommendationMajor Revision

---

## [Reviewer Report]

Reviewer name and names of any other individual's who aided in reviewer Jianbin Wang and James SimmonsDo you understand and agree to our policy of having open and named reviews, and having your review included with the published papers. (If no, please inform the editor that you cannot review this manuscript.)YesIs the language of sufficient quality?NoPlease add additional comments on language quality to clarify if needed
Please see commentsAre all data available and do they match the descriptions in the paper? YesAdditional CommentsThe evaluation of the data quality is lacking. Specifically, have the authors checked potential contaminations in the assembled genome, giving how much larger this genome compares to the genomes from other parasitic nematodes.Are the data and metadata consistent with relevant minimum information or reporting standards? See GigaDB checklists for examples <a href="http://gigadb.org/site/guide" target="_blank">http://gigadb.org/site/guide</a>YesAdditional CommentsIs the data acquisition clear, complete and methodologically sound?NoAdditional CommentsPlease see comments.Is there sufficient detail in the methods and data-processing steps to allow reproduction?NoAdditional CommentsPlease see commentsIs there sufficient data validation and statistical analyses of data quality? NoAdditional CommentsPlease see commentsIs the validation suitable for this type of data?NoAdditional CommentsPlease see commentsIs there sufficient information for others to reuse this dataset or integrate it with other data?YesAdditional CommentsAny Additional Overall Comments to the AuthorIn this manuscript, Hou et al. present a genome assembly for Aplectana chamaeleonis, a parasitic nematode that infects amphibians. They report a genome of ~1 Gb, most of which is composed of repetitive elements. This genome draft is significant as it is the first assembled for this or any Cosmocercidae species. It may provide insights into the evolution of the nematodes – if it is thoroughly compared to other nematode genomes. It may also allow for better species identification than previous morphological methods.  While the conclusions on genome size and composition described in the paper appear sound, there are many questions that go unanswered. The reasoning behind why this research was undertaken is not clear. What is the ecological or agricultural and economic impact of the species? How would the genome provide a better understanding of this species? More specific information is also needed to better understand the genome. How many chromosomes does this species have? Is there any cytology to help answer this question? Any notion of sex chromosome vs. autosome?  This genome is much bigger than most of the assembled parasitic nematodes. The author did not make any efforts to explain what might contribute to this. Could the big size due to contamination in the samples used? Judging from the images, it does not look very convincing to me how clean the sample was for the genomic DNA extraction. Overall, there is a lack of in-depth data analysis and comparison between this genome and many other available nematode genomes. About the overall presentation and organization of the manuscript, the context is often lacking from results. How do these results compare to related species? How does figure 4/the demographic history fit in to this story? A round of general proofreading needs to be done for grammar, punctuation, capitalization, italics, etc – see below for some specific examples. In the Abstract, the repeat content in the Ascaris genome is 72.45%, and the total length is more than 742 Mb. The math does not add up (1.1 Gb x 72.45% = 797 Mb). Or do you mean the Aplectana genome? Should say total length of repeats. Why is this “Ascaris” genome? Ascaris is a parasite that infects pigs and human. Some sentences need addressing/clarification: Page 1. “and their diversity is also very high, many of which are above the national second-level protected animals” – what is the significance of this/how are these ideas related?  Page 2. “Through the characteristics of the genome sequence, it shows that the genome is a highly continuous genome” – need to be more specific with metric and data. Page 4. “In addition, the enrichment of A. chamaeleonis genes in all metabolic pathways was found in twelve metabolic pathways.” – not sure what you are trying to say about the all or 12 pathways. Figure 1. - Images need scalebars. In A, what is the mat of material? For A, crop out area around the worm and enlarge the worm image. In B the worm is dark/shows little contrast or detail. In C, label which image is the head and which is the tail (or specify left vs. right in the legend text). The images in B and C look like they were taken using a cell phone pointed at a computer monitor – are there higher quality images? Table 1. – Why is the data in all four columns the exact same? What is the difference between each column? This appear to be a mistake when preparing the table. Very sloppy and unfortunate! Table 2 – Significant figures on the %s?. Is the “other” category needed (same for Fig2C)? Table 3 – Check text spacing (e.g. % in genome). Figure 3 – Recommend to redo the spacing of figures, increase size of text in each part of this figure. Need to refer to parts of figures in the body/text (Fig 3a vs. 3b vs. 3c). Can 3b be sorted from most number of genes to least? Figure 4 is not referenced in the body text. Consider merging Fig 4 with Fig 3. Figure 4 is lacking a description in the legend – what are the grey lines, definition of LGM? The x-axis scale and orientation are unintuitive – is the present on the left and the past on the right? Past should be on the left. Methods Genomic DNA was purification for Long-reads libraries preparation – should say purified What is the meaning of “The generation we used was 0.17” – what generation is this? and “the mutation rate was 9×10-9” needs units. The sentence “we used the pairwise sequentially Markovian coalescent (PSMC) model to estimate the effective population size of A. chamaeleonis within last million years.” should be moved to the section immediately after its current location.RecommendationMajor Revision